# From Genes to Healing: The Protective Mechanisms of *Poria cocos* Polysaccharide in Endometrial Health

**DOI:** 10.3390/cimb47030139

**Published:** 2025-02-21

**Authors:** Yongxi Huang, Pupu Yan, Jun Zhu, Yinuo Gong, Man Liu, Haishan Cheng, Tilin Yi, Fuxian Zhang, Xiaolin Yang, Yingbing Su, Liwei Guo

**Affiliations:** College of Animal Science and Technology, Yangtze University, Jingzhou 434025, China; 2022720956@yangtzeu.edu.cn (Y.H.); 2021720941@yangtzeu.edu.cn (P.Y.); 2023721037@yangtzeu.edu.cn (J.Z.); 15689567267@163.com (Y.G.); liuman94915@163.com (M.L.); chs041817@gmail.com (H.C.); yitilin@163.com (T.Y.); zhangfuxian99@163.com (F.Z.); yangxl@yangtzeu.edu.cn (X.Y.); su-yingbing@163.com (Y.S.)

**Keywords:** *Poria cocos* polysaccharide, endometritis, NFκB signaling pathway, apoptosis, transcriptome analysis

## Abstract

The aim of this study is to investigate the therapeutic effect of *Poria cocos* polysaccharide (PCP) on bovine endometritis. Initially, an inflammation model was induced using LPS-treated bovine endometrial epithelial cells (BEND) to identify the differentially expressed genes (DEGs) between the control and LPS groups by transcriptome sequencing, and GO functional annotation and KEGG enrichment analysis were performed. Subsequently, the mechanism of PCP treatment for endometritis was further evaluated using protein immunoblotting and real-time fluorescence quantitative analysis. Finally, the efficacy of PCP in treating endometritis was evaluated using a rat model of endometritis established with a mixed bacterial infection. The results show that transcriptome sequencing identified 4367 DEGs, with enrichment analysis highlighting the primary influences on the cell cycle and apoptosis signaling pathways. Following treatment of BEND with LPS resulted in cell apoptosis and inflammatory response. However, the introduction of PCP intervention significantly inhibited the progression of apoptosis and inflammation. Animal test results indicate that PCP significantly decreases the levels of serum inflammatory in rats suffering from endometritis and enhances antioxidant capacity. Furthermore, it effectively improved uterine swelling and tissue vacuolization caused by bacterial infection. These findings suggest that PCP could alleviate endometritis by modulating the inflammatory response and suppressing cell apoptosis. *Poria cocos* polysaccharides demonstrate significant potential for applications in immune modulation, anti-inflammatory responses, and antioxidant activities. Their high safety profile makes them suitable candidates as alternative therapeutic agents for the treatment of endometritis in the veterinary field.

## 1. Introduction

Persistent bacterial infection can result in a prolonged sub-health or non-health state of the uterus, which is a primary cause of infertility in cattle [1]. Endometritis, puerperal metritis, and pyometra are known as the three major diseases caused by intrauterine bacterial infection of bovine postpartum conditions [2]. Endometritis is a localized inflammation of the endometrium caused by *Escherichia coli*, *Staphylococcus aureus*, *Arcanobacterium pyogenes*, *Fusobacterium necrophorum*, and other pathogenic bacteria [3], which occurs frequently in perinatal cows [4], mainly manifested as vaginal purulent secretions; systemic symptoms are mild, but have a greater impact on the reproductive performance of cattle [5,6]. Currently, the primary treatment method for bovine endometritis is antibiotic therapy [7], which is cost-effective and efficacious, to some extent alleviating the outbreak of the disease and being widely applied in livestock and poultry farming. However, the presence of residual antibiotics and the emergence of antibiotic-resistant bacteria pose significant threats to human health and well-being [8]. Therefore, the development of environmentally friendly and efficient drugs for the treatment of bovine endometritis has emerged as a prominent focus in veterinary clinical research.

Since ancient times, traditional Chinese herbal medicine has been widely utilized for treating inflammation and related diseases. Based on a synthesis of modern scientific research, Chinese herbal medicine exhibits varying degrees of anti-inflammatory effects through multiple pathways and targets [9]. Particularly with the prohibition of adding antibiotics to livestock feed, the development of Chinese herbal medicine has entered a new phase [10]. The fungus known as *Poria*, belonging to the Polyporaceae family, has been recognized for its therapeutic properties as a traditional medicine [11]. It is documented to possess properties that include reducing water retention, dampness, tonifying the spleen, and calming the mind. Furthermore, existing research has revealed that *Poria cocos* possesses various therapeutic effects such as inhibiting acetylcholinesterase and displaying anti-inflammatory, antioxidant, and immune-regulating activities. These effects are highly dependent on the polysaccharide components of *Poria cocos* [12]. *Poria cocos* polysaccharides (PCP) have been shown to stimulate the activation of immune cells, thereby enhancing immune response [13]. In comparison to other therapeutic agents, whether synthetic or natural, Poria cocos exhibits distinct advantages. Notably, when contrasted with chemically synthesized pharmaceuticals, Poria cocos demonstrates reduced toxicity and enhanced biocompatibility, rendering it appropriate for prolonged administration [14]. Furthermore, relative to certain natural compounds, such as flavonoids or polyphenols, Poria cocos polysaccharides (PCP) exhibit more pronounced immunomodulatory and anti-tumor properties, particularly in terms of bolstering immune responses and suppressing tumor proliferation [11,15]. In addition, PCP can reduce oxidative stress and alleviate inflammatory responses by regulating the secretion of inflammatory factors and effectively clearing free radicals [16]. Overall, *Poria cocos* polysaccharides, as a natural compound, have significant advantages in terms of safety, versatility, and wide applicability, but there are several limitations that hinder its application, including low bioavailability, complex mechanisms of action, large individual variation, and scarce clinical data [17]. The existing literature on the utilization of PCP in the treatment of bovine endometritis is limited. This study evaluated the safety and efficacy of PCP and elucidated their mechanism of action.

This study investigated the feasibility of using PCP to treat endometritis through both in vivo and in vitro experiments. In vitro, experiments induced an inflammatory model in bovine endometrial epithelial cells (BEND) by stimulating them with LPS, and differential genes were identified using transcriptome sequencing. The mechanisms of action were validated in BEND using in vitro flow cytometry, real-time quantitative PCR, and Western blotting. Due to resource limitations and difficulties in constructing bovine endometritis models, LPS-induced bovine endometrial epithelial cells and in vivo mouse inflammation models have been used to explore the potential mechanisms of drug treatment of bovine endometritis [18,19]. Consequently, this study established an in vivo rat model of endometritis and subsequently verified the feasibility of using PCP to treat endometritis. This research will provide a theoretical foundation for the clinical development of PCP for the prevention and treatment of bovine endometritis.

## 2. Materials and Methods

### 2.1. Ethics Statement

This research was carried out in accordance with the Chinese Animal Welfare Guidelines and received approval from the Animal Ethics Committee at Yangtze University (YJ202346).

### 2.2. In Vitro Test

#### 2.2.1. Cell Culture and Viability Assay

BEND (Otwo Biotech, Shenzhen, China) were cultured in cell culture flasks using DMEM medium supplemented with 10% FPS (Fetal Bovine Serum) and 1% penicillin-streptomycin. The cell concentration was adjusted to 1 × 10^6^ cells/mL and was subsequently seeded into 96-well plates. Evenly mix *Poria cocos* polysaccharide into the DMEM culture medium to achieve a final concentration of 1 mg/mL. After sterilization at high temperature and pressure, serum-free DMEM culture medium was used for gradient dilution. Filter the lipopolysaccharide with a concentration of 1 mg/mL through a 22 μm filter membrane (BS-PES-22, Biosharp, Hefei, China), and then subject it to gradient dilution for subsequent experiments. After the cells adhered, different concentrations of PCP and LPS were added. After 23 h of incubation in a 37 °C incubator containing 5% carbon dioxide, 10 µL of CCK-8 (BS350A, Biosharp, Hefei, China) was added to each well, and incubation was continued for 1 h. Cell viability was assessed by measuring the optical density at 450 nm using a microplate reader (SpectraMax iD3, Molecular Devices, Shanghai, China).

#### 2.2.2. RNA Extraction and Transcriptome Sequencing

Total RNA was extracted from cellular samples, and the purity and integrity of the RNA were assessed utilizing a NanoDrop 2000 spectrophotometer (NanoDrop Technologies, Wilmington, DE, USA) and a Bioanalyzer 2100 system (Agilent Technologies, Santa Clara, CA, USA). Following the qualification of the samples, library construction was initiated. Initially, mRNA was isolated through the use of oligo(dT)-coated magnetic beads. The isolated RNA was subsequently fragmented with a fragmentation buffer, followed by reverse transcription employing random N6 primers, resulting in the synthesis of double-stranded cDNA. The cDNA was then repaired by incubation with a tail mix and RNA Index Adapters. The cDNA fragments generated in the preceding steps underwent PCR amplification, and the resultant products were purified using Ampure XP Beads (Beckman, Shanghai, China) to yield the final library. Upon completion of the library construction, preliminary quantification was conducted using Qubit 2.0, and the library was diluted to a concentration of 1.5 ng/mL. The sizes of the inserts within the library were analyzed using the Agilent 2100 BioAnalyzer (Agilent, Shanghai, China). The effective concentration was accurately determined through the RT-qPCR method to ensure the quality of the library. Once the library successfully passed the quality assessment, DNA nanoballs (DNBs) were prepared, loaded onto a sequencing chip, and sequenced using an MGI high-throughput sequencer (MGI, Shenzhen, China).

#### 2.2.3. Transcriptome Analysis

The raw reads underwent filtration using SOAPnuke software (v2.1.0) to generate clean reads. Subsequently, clean reads were aligned to the reference genome using HISTA2 (v2.1.0). Following this, post-quality control sequences were aligned to the reference transcript sequence using Bowtie2 (v2.3.5). Differential expression analysis was conducted using DESeq2 (V1.22.2), with a screening threshold set at false discovery rate (FDR) < 0.05 and log_2_FC (fold change) >1, or <−1 for significance determination. After selecting DEGs for further analysis, the distribution of these genes in the Gene Ontology (GO) was examined. Enrichment analysis was conducted using hypergeometric distribution, and GO terms with *Q*-Value ≤ 0.05 were selected as significantly enriched GO entries. The Kyoto Encyclopedia of Genes and Genomes (KEGG) was a central public database for pathways, annotated with KOBAS (V3.0). Pathways with *Q*-Value ≤ 0.05 were significantly enriched in DEGs. The pathway enrichment analysis was performed using R software (https://www.r-project.org/) in combination with self-written scripts and BH correction.

#### 2.2.4. Flow Cytometry Analysis

Cells infected with LPS were subjected to flow cytometry with membrane-associated protein-V or propidium iodide (PI) dual-staining to determine cell apoptosis. Data analysis was performed using flow Jo software (v10.8).

#### 2.2.5. Fluorescent Quantitative PCR (RT-qPCR) Detection of Relevant Genes

The total RNA was extracted using a TRIzol reagent kit (BS258A, Biosharp, Hefei, China), its concentration and purity were detected using an ultraviolet trace spectrophotometer (Thermo Fisher, Shanghai, China), and then reverse transcription for cDNA (RK20400, Abclonal, Wuhan, China) using a SYBR Green system for polymerase chain reaction amplification. We utilized a fluorescence quantitative analyzer for the assessment of the mRNA expression levels of the pertinent genes. RT-qPCR was performed using the following procedure: 95 °C for 3 min, followed by 30 cycles of 95 °C for 5 s and 60 °C for 30 s, with *GAPDH* as an internal reference. The 2^−ΔΔCt^ method was used to calculate the relative changes in mRNA. Primers from Primers were designed using Premier 5 software, with a product score greater than 98. The primer sequences are located in Table 1.

#### 2.2.6. Western Blot Analysis

After cultivation was completed, each group of cells was digested and collected. A RIPA lysis buffer (CW2333S, Cwbio, Beijing, China) containing protease inhibitors was added, and the mixture was placed on ice for 30 min. The protein concentration was adjusted using a BCA assay kit (PC0020, Solaibao, Beijing, China). Then, a 5× loading buffer was added, and the samples were denatured at 100 °C for 10 min to obtain cell protein samples. Equal amounts of protein (20 µg) were separated through 12% fast-gel electrophoresis (G2044, Servicebio, Wuhan, China) and then were transferred to a PVDF membrane that had been pre-treated with methanol. The membrane was sealed for 10 min using Western no-protein quick-sealing solution (BL1032B, Biosharp, Hefei, China). After rinsing with TBST for 30 s, the protein membrane was incubated with Caspase-3, Caspase-8, Caspase-9, BAX, BCL-2, TNF-α, NFκB, TLR4, Cyt-c, and β-actin rabbit monoclonal antibodies (Wanleibio, Wuhan, China) for 10 h, followed by TBST rinsing for 15 min. HRP-labeled sheep anti-rabbit IgG (SE134, Solaibao, Beijing, China) secondary antibody was added and incubated for 2 h. The membrane was then rinsed with TBST for 15 min. ECL luminescence reagent (MA0186, Meilunbio, Shanghai, China) was used for color development, and the results were observed through a chemiluminescence image analysis system. ImageJ software was used to analyze the grayscale values of the bands. Using β-actin as an internal reference, three replicates were set for each group.

### 2.3. The Efficacy Experiment of PCP in Treating Endometritis

#### 2.3.1. Source of Polysaccharide and Ciprofloxacin Hydrochloride

PCP was acquired through established methodologies [20] and stored within the Laboratory of Chinese Veterinary Medicine at Yangtze University. Ciprofloxacin hydrochloride was bought from Changtai Animal Pharmaceutical Co., Ltd., Yuncheng, China.

#### 2.3.2. Experimental Design and Management

A total of 36 healthy rats (Wuhan Rat Bayley Biotechnology Co., Ltd., Wuhan, China) were housed in separate cages. The rat enclosures were situated in a well-ventilated environment maintained at a temperature range of 21–26 °C, with a relative humidity of 40 ± 10% and appropriate light levels. Daily monitoring of food and water consumption was conducted, and drinking water was replenished every 8 h. After 7 days of adaptive feeding, 36 rats were randomly divided into 6 groups: blank group (without any processing), model group, Ciprofloxacin hydrochloride group (20 mg/kg Antibiotics), low dose of PCP group (L-PCP, 100 mg/kg PCP), medium dose of PCP group (M-PCP, 200 mg/kg PCP), and high dose of PCP group (H-PCP, 400 mg/kg PCP). In order to create a model of endometritis [21], rats were injected with 3% acetic acid to stimulate the uterus for 24 h. Subsequently, a suspension of *Staphylococcus aureus* and *Escherichia coli* (1 × 10^9^ CFU/mL, 1:2) was injected through the vaginal opening for three consecutive days, excluding the control group. Following the successful establishment of the endometritis model, the drug dosage treatment was administered consecutively for a duration of seven days in adherence to the specified protocol. After the last dose administration for 48 h, blood samples were collected from the rats’ eyeballs as soon as possible, immediately transferred to the laboratory, allowed to clot at 37 °C for 1 h, and then centrifuged at 4000× *g* for 15 min to separate the serum. The serum samples were stored at −20 °C for further analysis of inflammatory factors. Subsequently, the rats were euthanized, their body weights were recorded, and uterine, liver, and spleen tissues were collected and stored at −80 °C for further analysis.

#### 2.3.3. Organ Index, Liver Antioxidant Levels, and Inflammatory Factors

The collected rat uterus, liver, and spleen tissues were weighed to calculate the organ indices. The rat liver was accurately weighed, and 9 times the volume of physiological saline was added, followed by thorough grinding. The suspension was used to detect the content of MDA (A003-1), GSH (A005-1-2), SOD (A001-3-2), and CAT (A007-2-1) using kits (Nanjing Jiancheng, Nanjing, China) according to the operation instructions. The serum levels of IL-4 (SEKR-004), IL-6 (SEKR-005), IL-10 (SEKR-006), and TNF-α (SEKR-009) were measured using Sandwich ELISA kits (Solaibao, Beijing, China).

#### 2.3.4. Effect of PCP on Inflammatory Factors in Endometritis Rats

The RNA extraction from uterine tissue was carried out using the Trizol reagent kit (BS258A, Biosharp, Hefei, China). The concentration and purity of the total RNA were assessed using an ultraviolet microspectrophotometer, followed by reverse transcription into cDNA (RK20400, Abclonal, Wuhan, China). Polymerase chain reaction (PCR) was conducted utilizing the SYBR Green system. GAPDH served as the internal control and the mRNA expression levels of relevant inflammatory factors were analyzed and calculated using the 2^−ΔΔCt^ method. The primers were designed with Premier 5 software, ensuring a product score exceeding 98. The primer sequences’ positions are detailed in Table 2.

#### 2.3.5. Pathological Changes in Uterine Tissue

The freshly collected uterine tissue was fixed in 4% paraformaldehyde, dehydrated in an ethanol gradient, made transparent in xylene, and embedded in paraffin. Then, 5 μm sections of each sample were stained with hematoxylin and eosin (H&E). The pathological changes in the uterine tissues were observed under an XD30A-RFL microscope (Sunny Optical Technology Co., Ltd., Yuyao, China). The Image Pro-Plus (IPP) software (V6.0, Media Cybernetics, Rockville, MA, USA) was utilized to quantify the area of the inflammatory cell infiltration region, thereby facilitating a comparative analysis of inflammation severity across various groups.

### 2.4. Statistical Analysis

A one-way ANOVA was performed utilizing SPSS software (version 27.0, SPSS Inc., Chicago, IL, USA), with Duncan’s multiple range test employed to evaluate the significance of differences among means. The statistical significance of these differences is denoted by the presence of letters a–e within a column; distinct letters signify significant differences between means (*p* < 0.05). Data visualization was executed using GraphPad Prism 9.5 (GraphPad Inc., La Jolla, CA, USA), while data measurements were carried out with ImageJ software (version 1.8). All experimental data were independently replicated a minimum of three times, and the results are presented as mean ± standard deviation (SD).

## 3. Results

### 3.1. Cell Viability Assay

The impact of PCP on the growth of BEND is illustrated in Figure 1A. In comparison to the control group, no statistically significant variances were observed when exposed to PCP concentrations of 5 µg/mL, 10 µg/mL, and 15 µg/mL (*p* > 0.05). However, notable distinctions were evident between the two groups when subjected to PCP concentrations of 20 µg/mL (*p* < 0.05). Therefore, concentrations of 5 µg/mL, 10 µg/mL, and 15 µg/mL of PCP were chosen for the subsequent experiment. Figure 1B illustrates the impact of LPS on cellular viability. A notable disparity in cell viability was observed at a concentration of 10 µg/mL of LPS in comparison with the control group (*p* < 0.05). To better induce apoptosis and inflammation models, a consistent concentration of 40 μg/mL LPS was used during the experiment.

### 3.2. Transcriptome Data

A comparative analysis of transcriptome sequencing outcomes for the control and LPS groups is presented in Table 3. Each sample yielded over 2.5 million sequences and more than 6 GB of clean data. The average percentages of Q20 and Q30 bases were 99.09% and 96.71%, respectively, with an average GC content of approximately 51.04% across the samples. These results validate the robust quality of the sequencing data, thereby underpinning the reliability of subsequent analyses. Moreover, the association among samples indicates the similarity of various treated samples in terms of their expression levels. A correlation coefficient closer to 1 signifies a greater resemblance between samples. As depicted in Figure 2A, all correlation coefficients exceeded 0.97. Additionally, the comparison of overall expression profiles among samples was conducted through expression density profiles and box plots, as demonstrated in Figure 2B,C. These findings further underscored the heightened similarity observed among the samples.

### 3.3. Differences in Gene Identification

Through the examination of sequencing data, genes exhibiting significantly different expression levels (*p* < 0.05) were identified as Differentially Expressed Genes (DEGs). MA and volcano plots were generated to visually represent the False Discovery Rate (FDR) distributions and Fold Change (FC) values of all genes across the two groups, as depicted in Figure 2D,E. Furthermore, a cluster heatmap of DEGs was constructed to illustrate the varying gene expression levels and patterns among the six samples collectively, as illustrated in Figure 2F. These results show that a total of 4367 DEGs were identified between the control and LPS groups, of which 1819 were upregulated and 2548 were downregulated.

### 3.4. GO and KEGG Enrichment Analysis

To further investigate the roles of the DEGs, enrichment analyses were conducted using the Gene Ontology (GO) and Kyoto Encyclopedia of Genes and Genomes (KEGG) databases. Figure 3A illustrated the enrichment of 3554 GO terms, encompassing categories such as cellular processes, metabolism, and biological functions. The KEGG enrichment results presented in Figure 3B delineate 250 signaling pathways that are categorized into five branches, including transport, catalysis, cell growth, and death. Notably, the upregulated DEGs were associated with apoptosis, whereas the downregulated DEGs corresponded to the cell cycle, as shown in Figure 3C,D.

### 3.5. BEND Cell Apoptosis

The results of the flow cytometry detection of cell apoptosis are shown in Figure 4A. It was observed that apoptosis was significantly increased in the LPS group compared to the control group. The administration of PCP significantly inhibited apoptosis at all doses (*p* < 0.05). Moreover, there was no significant difference in the anti-apoptotic effect between 5 µg/mL and 15 µg/mL PCP (*p* > 0.05). Furthermore, this research reveals that the expression levels of genes associated with apoptosis and inflammation were changed following stimulation with LPS. The gene expression level pertaining to cell apoptosis is illustrated in Figure 4B–I, including *TNF-α* (Figure 4B), *FADD* (Figure 4C), *TRADD* (Figure 4D), *BAX* (Figure 4E), *Caspase-3* (Figure 4F), *Caspase-8* (Figure 4G), and *Caspase-9* (Figure 4H), displayed a significant increase in the LPS group compared to the control group (*p* < 0.05) while showing a significant decrease in *BCL-2* (Figure 4I) (*p* < 0.05). Following PCP intervention, the mRNA expression levels of these genes were reinstated, particularly in the 10 µg/mL and 15 µg/mL PCP groups (*p* < 0.05). Upon exposure to LPS, the expression levels of inflammatory factors are shown in Figure 5. There was a notable increase in the mRNA expression levels of *TLR4* (Figure 5A), *NFκB* (Figure 5B), *IL-1β* (Figure 5C), and *IL-6* (Figure 5D), while the mRNA expression level of *IL-10* (Figure 5E) exhibited a significant decrease compared to the control group (*p* < 0.05). Treatment with PCP led to the restoration of the mRNA expression levels of these genes, particularly in the 15 μg/mL PCP group (*p* < 0.05).

### 3.6. Fluorescent Quantitative PCR Detection of Relevant Genes

The results in Figure 6 demonstrate the genes’ mRNA expression associated with the cell cycle and cell apoptosis, thereby confirming the precision of transcriptome sequencing. The mRNA expression levels of key genes such as *CDK4* (Figure 6A), *CCNA2* (Figure 6B), *CCND1* (Figure 6C), and *CDK6* (Figure 6D), as well as pivotal genes in the apoptosis signaling pathway, namely *Caspase-4* (Figure 6E) and *BIRC3* (Figure 6F), were quantitatively assessed. In the LPS group, a notable decrease in the mRNA expression levels of *CDK4*, *CCNA2*, and *CCND1* was observed compared to the control group (*p* < 0.05). Subsequent treatment with PCP led to a gradual increase in these genes, particularly in the 15 µg/mL PCP group, where the mRNA expression levels of *CDK4*, *CCNA2*, and *CCND1* were significantly elevated (*p* < 0.05). Conversely, the mRNA expression level of *CDK6* in the LPS group exhibited a significant increase compared to the control group (*p* < 0.05), with the 15 µg/mL PCP group showing a marked decrease relative to the LPS group (*p* < 0.05). Furthermore, the mRNA expression levels of *Caspase-4* and *BIRC3* were significantly elevated in the LPS group (*p* < 0.05), but, following PCP intervention, their expression levels notably decreased (*p* < 0.05).

### 3.7. The Protein Expression Levels of NFκB and Apoptosis Signaling Pathways

This study assessed the expression levels of proteins associated with inflammation and apoptosis signaling pathways, as depicted in Figure 7. Following exposure to LPS, the protein expression levels of TLR4 (Figure 7A), NFκB (Figure 7B), BAX (Figure 7C), TNF-α (Figure 7D), Caspase-3 (Figure 7E), Caspase-9 (Figure 7F), Cyt-c (Figure 7G), and Caspase-8 (Figure 7H) exhibited a significant increase compared to the control group (*p* < 0.05). Treatment with PCP effectively reversed this situation. Notably, the expression of the anti-apoptotic protein BCL-2 (Figure 7I) was significantly reduced in the LPS group compared to the control group (*p* < 0.05). Post PCP intervention, the protein level of BCL-2 gradually rose, with a significant increase observed in the 10 µg/mL and 15 µg/mL groups compared to the control group (*p* < 0.05).

### 3.8. Results of Organ Indexes

After bacterial infection, the organ index of rats is shown in Table 4. Compared to the blank group, the uterine index in the model group significantly increased (*p* < 0.05). The uterine index of rats displayed a gradual decrease with escalating doses of PCP, with no significant variance in comparison with the antibiotic-treated group (*p* > 0.05). In the H-PCP group and antibiotic-treated group, the spleen index exhibited a statistically significant decrease compared to the model group (*p* < 0.05). Conversely, the results in the L-PCP and M-PCP groups did not demonstrate a significant difference in spleen index compared to the model group (*p* > 0.05). The liver index did not show any significant differences among all groups (*p* > 0.05). Notably, an increase in PCP drug concentration correlated with a dose-dependent decline in the liver index.

### 3.9. Results of Antioxidant Level in Liver and Inflammatory Factors

The MDA content in the rat liver is shown in Figure 8A. Compared to the blank group, the model group showed significant accumulation of MDA in the liver. Treatment with both PCP and antibiotics could significantly reduce the MDA content. Notably, compared to the antibiotic group, the high dose of PCP had a more pronounced effect in reducing MDA content. Subsequent treatment with PCP and antibiotics resulted in a significant decrease in MDA content compared to the model group (*p* < 0.05), eventually returning to baseline levels. In Figure 8B, the SOD content in the model group was significantly lower than in the control group (*p* < 0.05). The H-PCP and M-PCP groups exhibited a more favorable increase in SOD content compared to the model group. Although the PCP low-dose group and antibiotic group did not reach optimal levels, there was no significant difference compared to the control group. Figure 8C,D displayed the changes in GSH and CAT levels. In comparison to the control group, a notable decrease in GSH content was observed in the model group (*p* < 0.05). Conversely, the medium and high doses of PCP significantly increased in GSH content compared to the model group (*p* < 0.05). The CAT content in both the drug treatment and antibiotic groups exhibited significant differences from that in the model group (*p* < 0.05). The high dose of PCP and antibiotic treatment demonstrated the most effective outcomes. As shown in Figure 8E, compared to the control group, the IL-6 levels in the model group were significantly elevated (*p* < 0.05). Although there was still a significant difference after treatment with PCP (*p* < 0.05), as the dose of PCP increased, the IL-6 levels gradually decreased and were not significantly different from the antibiotic group (*p* > 0.05). Additionally, as shown in Figure 8F, the levels of TNF-α in the antibiotic group and PCP groups were both higher than those in the control group (*p* < 0.05). It is worth noting that the levels of TNF-α showed a decreasing trend in dose dependency. As shown in Figure 8G,H, compared to the blank group, the IL-4 and IL-10 levels in the model group were significantly decreased, while the IL-4 and IL-10 levels in the high-dose PCP group were significantly increased (*p* < 0.05). There was no statistically significant difference in IL-4 and IL-10 levels in the medium-dose PCP group and the antibiotic group (*p* > 0.05). In addition, although the levels of IL-10 in the L-PCP and M-PCP groups showed a statistically significant difference compared to the blank group (*p* < 0.05), there was a trend in dose-dependent recovery.

### 3.10. Results of mRNA Expression of Related Inflammatory Factors

The results of mRNA expression of related inflammatory factors are illustrated in Figure 2. Following a mixed bacterial infection, the experimental groups showed significantly higher levels of *NFκB* (Figure 9A), *TLR4* (Figure 9B), *IL-6* (Figure 9C), *TNF-α* (Figure 9D), and *IL-1β* (Figure 9E) compared to the control group (*p* < 0.05). In H-PCP and M-PCP groups, the inflammatory markers exhibited a more pronounced reduction (*p* < 0.05), and the group receiving antibiotics also showed a significant improvement in inflammatory markers. Additionally, post-bacterial infection, the expression of *IL-5* (Figure 9F) was notably decreased compared to the control group (*p* < 0.05). Compared to the model group, the middle- and high-dose groups and the antibiotic group had a significant increase in the expression level (*p* < 0.05), and there was no significant difference between the antibiotic group and the blank group (*p* > 0.05).

### 3.11. Pathological Observation of Uterine Tissue in Rats

The pathological section of rat uterine tissue is shown in Figure 10. In the control group, the uterus was intact in shape and structure, with no inflammatory cell infiltration and dense cell arrangement. Conversely, the model group showed increased inflammatory cell infiltration, tissue swelling, expanded interstitial space, and pronounced tissue vacuolization. Following drug treatment, there was a notable improvement in the pathological features, except in the low-dose group, which showed a decrease in vacuolization. The middle- and high-dose groups demonstrated a clear uterine structure with minimal vacuolization. The antibiotic group exhibited the most effective therapeutic outcome, characterized by intact uterine tissue and regular cell arrangement. The relative quantitative results are shown in Table 5. Compared to the blank group, the rate of inflammatory cell infiltration in the model group was 26.76%, a finding that reached statistical significance (*p* < 0.05). When compared to the model group, the infiltration of inflammatory cells within the uterine tissue of rats in the M-PCP, H-PCP, and antibiotic groups was reduced to 15.87%, 13.13%, and 15.63%, respectively, with all differences also achieving statistical significance (*p* < 0.05).

## 4. Discussion

Endometritis in cattle during the delivery or postpartum periods is typically attributed to microbial infections, predominantly involving purulent and Gram-negative anaerobic bacteria contamination of the genital tract [22]. The clinical signs include heightened vaginal discharge and swelling and congestion of the cervix, leading to adverse impacts on the well-being and reproductive efficiency of cattle, resulting in substantial financial repercussions [23,24]. Inflammation is the hallmark response of the innate immune system when foreign substances or pathogens are detected [25]. Mild inflammation helps the uterus clear invading pathogens and initiates tissue healing and repair, but excessive inflammation can damage the endometrial tissue, leading to a decrease in the reproductive performance of cattle.

Existing studies have shown that traditional Chinese medicine has a good therapeutic effect on bovine endometritis [26,27]. As a widely recognized traditional medicinal fungus, *Poria cocos* holds significant medicinal and nutritional importance, such as anti-inflammatory, anti-oxidation, spleen and liver protection, diuresis, immune enhancement, and regulation of intestinal microorganisms [28,29,30]. Polysaccharides are one of the main bioactive components of *Poria cocos* and have been widely used in the livestock and poultry industries. Does *Poria cocos* polysaccharide have a therapeutic effect on bovine endometritis? This remains to be proven as an alternative treatment for endometritis. In this study, an in vitro inflammation cell model utilizing BEND stimulated by LPS was employed to investigate the mechanisms through which PCP mitigates bovine endometritis. However, owing to constraints in resources and the challenges associated with developing a bovine endometritis model, we opted to create a rat animal disease model to evaluate the therapeutic efficacy of PCP in the treatment of endometritis.

Advancements in technology have enabled the utilization of microarray and bioinformatics methodologies for transcriptome analysis, facilitating the identification of common biomarkers across various diseases and providing valuable insights into the underlying mechanisms of disease pathogenesis [31,32,33]. To explore PCP’s underlying mechanisms of treating bovine endometritis, transcriptome sequencing was performed on normal BEND- and LPS-induced inflammatory injury cell models to analyze the DEGs firstly in this study. Based on the above results, GO and KEGG enrichment analyses were conducted to investigate the reaction mechanisms of biological signals and energy metabolisms under pathological conditions, as well as the functional connections between genes. Consequently, 4367 differentially expressed genes obtained from transcriptome sequencing were found to be mainly involved in the signaling pathways related to apoptosis, cell cycle, and inflammation using enrichment analysis. This result was consistent with previous reports [34,35,36].

An increasing body of research suggests that cellular apoptosis and inflammatory responses are closely linked, indicating a potential therapeutic approach for inflammatory diseases [37]. In this study, flow cytometry results showed that LPS stimulation triggered significant cell apoptosis, but under PCP intervention, apoptosis was significantly reduced. This result further verified the anti-inflammatory effect of PCP. Transcriptome analysis has demonstrated that DEGs predominantly influence the cell cycle and apoptosis signaling pathways. Subsequent investigations have highlighted the significant roles of genes such as *BIRC3*, *Caspase-4*, and *CCNA2* in these processes. In the process of apoptosis, the baculovirus inhibitor containing the apoptotic repeat (BIRC) gene may signal inflammatory transcription factors, nuclear factor-κB (NF-κB), and protection from cell death [38]. BIRC3 (baculoviral IAP repeat containing 3) plays an important role in the process of apoptosis. As a member of the IAP protein family, BIRC3 exerts its anti-apoptotic function by interacting with tumor necrosis factor receptor-associated factors TRAF1 and TRAF2 [39]. Moreover, Monica et al. discovered that an inflammatory milieu facilitates disease progression and the infiltration of pathogens. The inflammatory cytokine *IL-1β* can enhance the expression of BIRC3, thereby augmenting cellular resistance [40]. Concurrently, Kelly et al. demonstrated that caspase-4-specific siRNA effectively inhibits *IL-1β* production, validating the reliance of endometrial cell inflammasomes on activated caspase-4 [41]. The findings of our study reveal a notable upregulation in the expression levels of *TLR4*, *NFκB*, *IL-1β*, *IL-6*, *Caspase-4*, and *BIRC3*, alongside a significant decrease in *IL-10* expression in BEND following LPS stimulation. These alterations suggested the activation of apoptotic pathways and an inflammatory reaction. Subsequent treatment with PCP restored cellular expression levels to baseline, indicating that PCP had a good anti-inflammatory effect by regulating the apoptotic pathways. Cyclin-dependent kinase (CDK) is a pivotal regulator in the control of the cell cycle in mammals. *CDK4* and *CDK6* oversee the G1/S restriction checkpoint by interacting with cyclin D proteins upon stimulation by growth factor signals [42]. In the current investigation, it was observed that the expression of *CDK4* was reduced while *CDK6* expression was increased following LPS infection. Subsequent administration of PCP resulted in the restoration of both CDK4 and CDK6 levels in a manner dependent on the dosage of the drug, potentially implicating a role in cell proliferation. *CCND1* and *CCNA2* belong to the cyclin family and mainly promote cell proliferation and regulate the cell cycle [43,44]. Our results found that PCP restored the expression levels of *CCND1* and *CCNA2*, which were reduced in the LPS-induced model. Caspases are a family of cysteine proteases that play a crucial role in the execution of apoptosis and the regulation of inflammation [45]. Fu et al. reported that extracted polysaccharide can suppress the chicken embryos’ raw muscle cells apoptosis-related gene expression of Caspase-1 and Caspase-3 induced by LPS [46]. The BCL-2 family of proteins modify mitochondria-dependent cell apoptosis, acting as anti-apoptotic proteins. Cytochrome c (Cyt-c) release from the mitochondrial membrane to the cytoplasm mediates cell death, which is mediated by the abnormal expression of BCL-2 family genes [47]. The BCL-2/BAX ratio-mediated apoptosis signaling pathway can trigger Caspase-9 activation, which in turn modulates Caspase-3 activation, leading to cell apoptosis. PCP significantly increased the gene and protein expression of the anti-apoptotic protein BCL-2 while inhibiting the expression of the pro-apoptotic protein BAX. This resulted in an increased BCL-2/BAX ratio which suppressed the mitochondrial-dependent apoptotic signaling pathway. Caspase-3 plays a pivotal role in cell apoptosis by cleaving essential cellular proteins, initiating a cascade of events resulting in cell death. Studies have shown that LPS infection enhanced the expression of Caspase-3, Caspase-8, Caspase-9, and BAX genes and proteins while reducing the levels of BCL-2 gene expression and protein abundance. Conversely, treatment with PCP markedly elevated BCL-2 gene and protein expression, suppressed the expression of Caspase-3, Caspase-8, Caspase-9, and BAX. Moreover, key upstream regulatory genes of cell apoptosis such as TNF-α, FADD, and TRADD were upregulated following LPS induction, but their expression was significantly inhibited by PCP treatment. Furthermore, PCP markedly decreased the protein expression of Cyt-c and impeded the translocation of Cyt-c from the mitochondrial membrane to the cytoplasm, effectively obstructing the mitochondrial-mediated apoptotic pathway. Overall, PCP demonstrates anti-apoptotic properties through the modulation of BCL-2 family proteins, the inhibition of the Caspase cascade, the reduction in Cyt-c release, and the downregulation of upstream apoptosis regulatory genes.

Toll-like receptors (TLRs) are transmembrane receptors crucial for the innate immune response as they detect specific pathogen-associated molecular patterns (PAMPs) from invading bacteria [48,49]. Lipoteichoic acid (LTA) and Lipopolysaccharides (LPS) are significant components of PAMPs that trigger an inflammatory reaction in vivo [50], leading to the release of inflammatory cytokines such as IL-1β, IL-6, and TNF-α [51]. These molecules can activate the immune system by initiating a signaling cascade within cells upon TLR-PAMP interaction. Notably, nuclear factor kappa B (NF-κB) plays a pivotal role in inflammation, with NFκB p65 and IκB-α undergoing phosphorylation post-activation, leading to p65 translocation into the nucleus and the subsequent production of proinflammatory mediators [52]. In this research, with the treatment with PCP and antibiotics, the serum levels of proinflammatory factors TNF-α and IL-6 were significantly decreased, but the levels of anti-inflammatory cytokines IL-10 and IL-4 were elevated, leading to a reduction in the body’s inflammatory response. Concurrently, the expression of inflammation-associated cytokines, including TLR4, NFκB, and IL-6, within uterine tissue is predominantly modulated by the administration of PCP and antibiotics. Furthermore, the type 2 cytokine interleukin-5 is crucial for the activation of both innate and adaptive immune responses, the preservation of metabolic homeostasis, and the facilitation of tissue repair [53]. Following treatment with PCP and antibiotics, there is a marked increase in the expression levels of IL-5. These findings indicate that PCP inhibits inflammation through the TLR4/NFκB signaling pathway, thereby modulating the expression of relevant genes and effectively alleviating endometrial inflammation induced by mixed bacteria in rats.

Inflammation and oxidative stress are intricately linked processes that play a significant role in the body’s defense against infectious agents [54]. When the body is exposed to harmful stimuli, there is an excessive production of highly reactive molecules such as active oxygen free radicals (ROS) and reactive nitrogen free radicals (RNS) [55,56]. The degree in oxidation exceeds the removal of oxides, leading to a severe imbalance between the oxidation system and the antioxidant system, resulting in tissue damage [57]. In this study, histopathological analyses indicated that both PCP and antibiotics significantly mitigated bacterial-induced damage to uterine tissue. Furthermore, following the administration of PCP and antibiotic treatment, there was a notable reduction in the accumulation of MDA in the liver. Concurrently, the expression levels of CAT, GSH, and SOD were elevated, leading to a decrease in oxidative stress induced by bacterial activity, contributing to an enhancement in the organism’s antioxidant capacity and its ability to repair tissue. This intervention resulted in a decrease in the infiltration of inflammatory cells within the affected tissue. Notably, high doses of PCP demonstrated superior therapeutic efficacy in the treatment of endometritis compared to antibiotics. The polysaccharides possess inherent, gentle, and sustained benefits in diminishing inflammation, enhancing tissue repair, and preventing recurrence, thus effectively averting disease relapse and rendering them suitable for prolonged administration with minimal adverse effects. Conversely, while antibiotics are highly effective in managing acute infections, their long-term use may lead to the development of resistance and associated side effects. The concurrent application of both treatment modalities may yield a synergistic effect in certain scenarios. However, the choice of treatment should be tailored to the specific clinical context.

## 5. Conclusions

In this study, PCP had the potential to treat endometritis. PCP regulates the NFκB and apoptosis signaling pathways to exert a protective effect against inflammatory damage. These results emphasized the role of PCP in alleviating LPS-induced inflammatory damage in BEND, providing a theoretical basis for further research on the in vivo application of PCP in treating bovine endometritis.

## Figures and Tables

**Figure 1 cimb-47-00139-f001:**
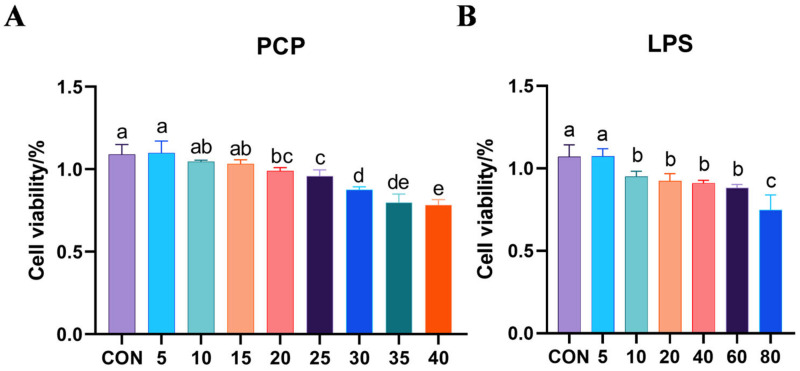
Cell vitality test results. (**A**) Cell proliferation situation, with PCP concentration on the horizontal axis and cell viability on the vertical axis. (**B**) The impact of different concentrations of LPS stimulation on the vitality of BEND: the horizontal axis represents the concentration of LPS and the vertical axis represents cell viability. All experiments were repeated more than three times and are presented as mean ± SD (significant differences (*p* < 0.05) between groups with different superscripts a, b, c, d and e).

**Figure 2 cimb-47-00139-f002:**
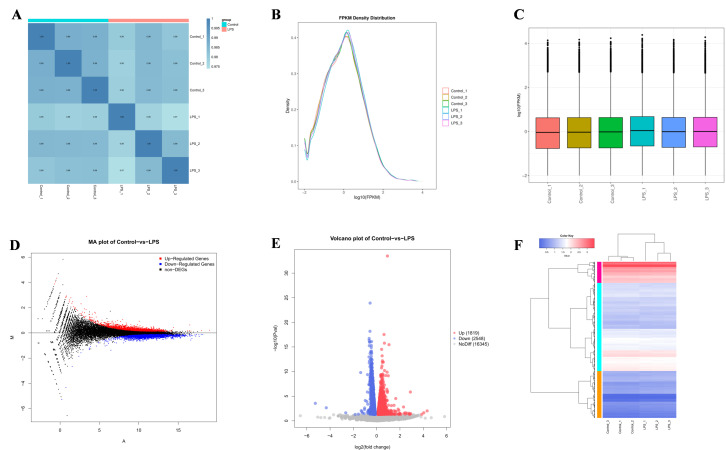
BEND LPS infects the sequencing results. (**A**) Correlation between the level of gene expression of heat maps in all samples: red represents the model group and blue represents the blank group. (**B**) Different samples of gene transcription/this Fpkm density figure; different colors represent different samples. (**C**) Fpkm box plots of genes/transcripts from different samples, with different colors representing different samples. (**D**) Differential gene Ma plot, with black representing normal genes, red representing upregulated genes, and blue representing downregulated genes. (**E**) Volcano map of DEGs, with black representing normal genes, red representing upregulated genes, and blue representing downregulated genes. (**F**) Heatmap of differential gene expression, with the abscissa representing samples and the ordinate representing kinship. Different colors distinguish the distribution of gene expression.

**Figure 3 cimb-47-00139-f003:**
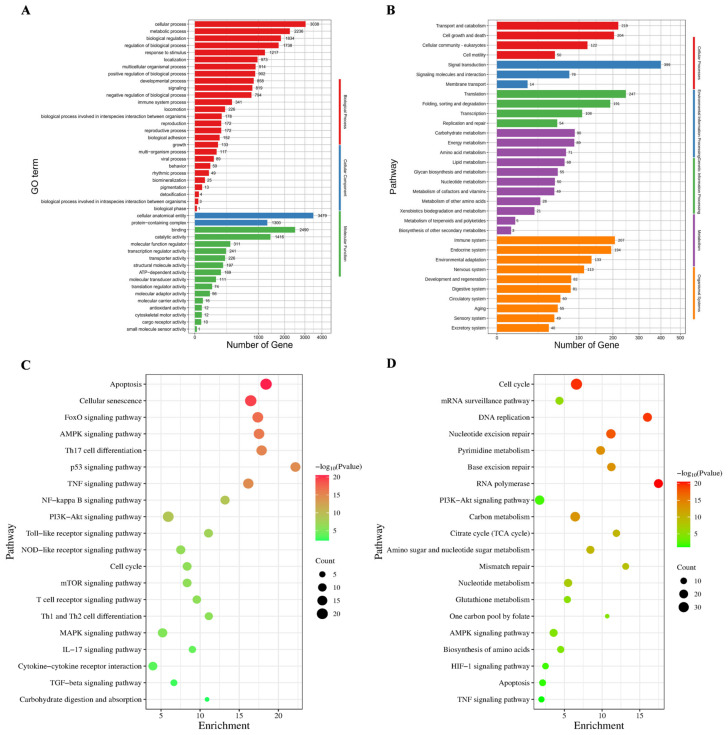
GO functional annotation and KEGG enrichment analysis based on DEGs. (**A**) The horizontal axis represents the enrichment factor, the vertical axis represents the GO project names, and the Q values are sorted from small to large. The size of the columns represents the number of DEGs in the GO project. (**B**) The vertical axis represents the names of KEGG metabolic pathways and the horizontal axis represents the number of genes annotated in that pathway. Genes are classified into five branches based on the involved KEGG metabolic pathways: cellular processes, environmental information processing, genetic information processing, metabolism, and organismal systems. (**C**) Enrichment of KEGG signaling pathways associated with upregulated genes. (**D**) Enrichment of KEGG signaling pathways associated with downregulated genes.

**Figure 4 cimb-47-00139-f004:**
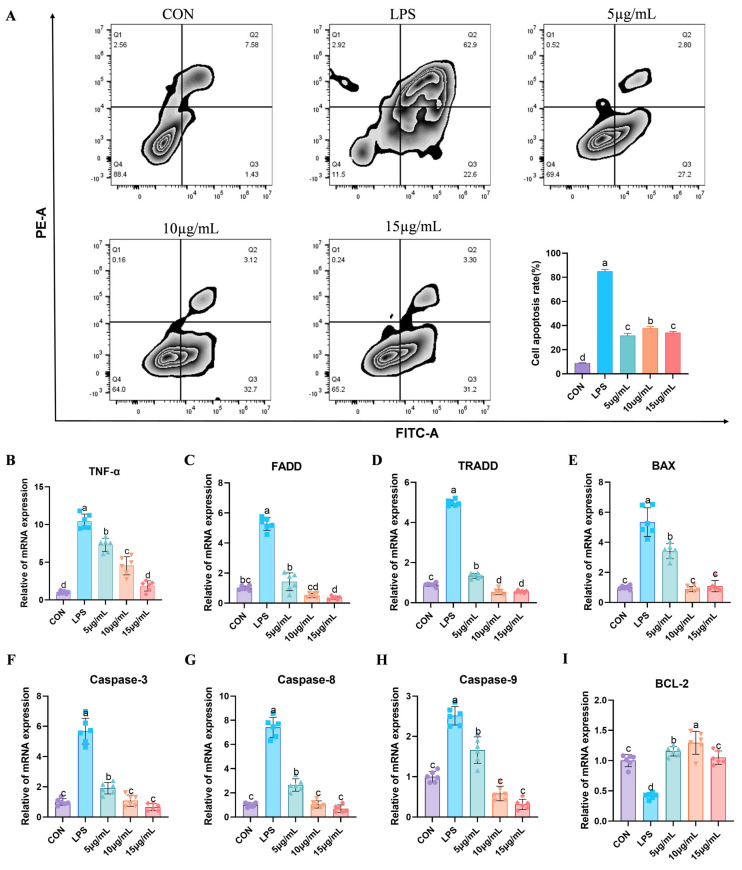
Flow cytometry detection of cell apoptosis and the expression levels of apoptosis-related genes. (**A**) BEND cell apoptosis, (**B**) *TNF-α* mRNA expression, (**C**) *FADD* mRNA expression, (**D**) *TRADD* mRNA expression, (**E**) *BAX* mRNA expression, (**F**) *Caspase-3* mRNA expression, (**G**) *Caspase-8* mRNA expression, (**H**) *Caspase-9* mRNA expression, (**I**) *BCL-2* mRNA expression. All experiments were repeated more than three times and are presented as mean ± SD (significant differences (*p* < 0.05) between groups with different superscripts a, b, c, and d).

**Figure 5 cimb-47-00139-f005:**
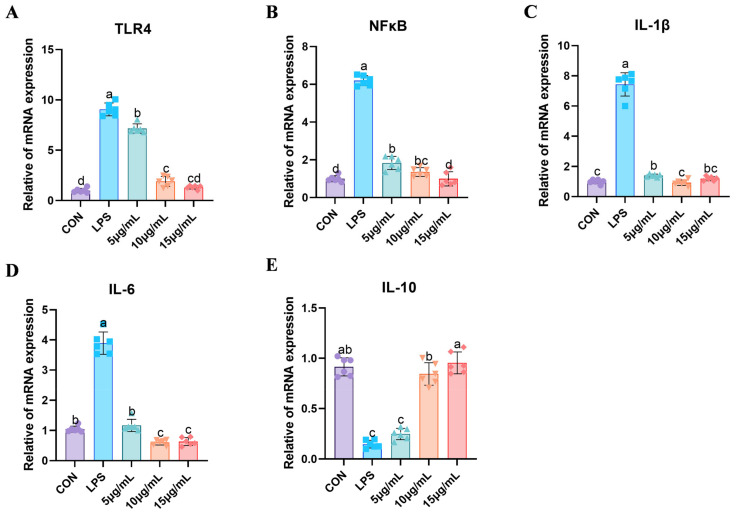
Detect the expression levels of inflammatory-related gene mRNA using the RT-qPCR method. (**A**) *TLR4* mRNA expression, (**B**) *NFκB* mRNA expression, (**C**) *IL-1β* mRNA expression, (**D**) *IL-6* mRNA expression, (**E**) *IL-10* mRNA expression. All experiments were repeated more than three times and are presented as mean ± SD (significant differences (*p* < 0.05) between groups with different superscripts a, b, c, and d).

**Figure 6 cimb-47-00139-f006:**
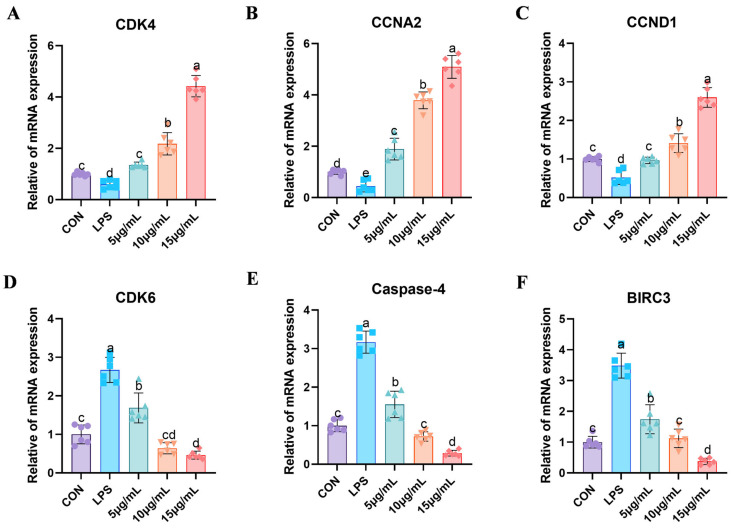
Fluorescence quantitative validation of transcriptome sequencing results. (**A**) *CDK4* mRNA expression, (**B**) *CCNA2* mRNA expression, (**C**) *CCND1* mRNA expression, (**D**) *CDK6* mRNA expression, (**E**) *Caspase-4* mRNA expression, (**F**) *BIRC3* mRNA expression. All experiments were repeated more than three times and are presented as mean ± SD (significant differences (*p* < 0.05) between groups with different superscripts a, b, c, d and e).

**Figure 7 cimb-47-00139-f007:**
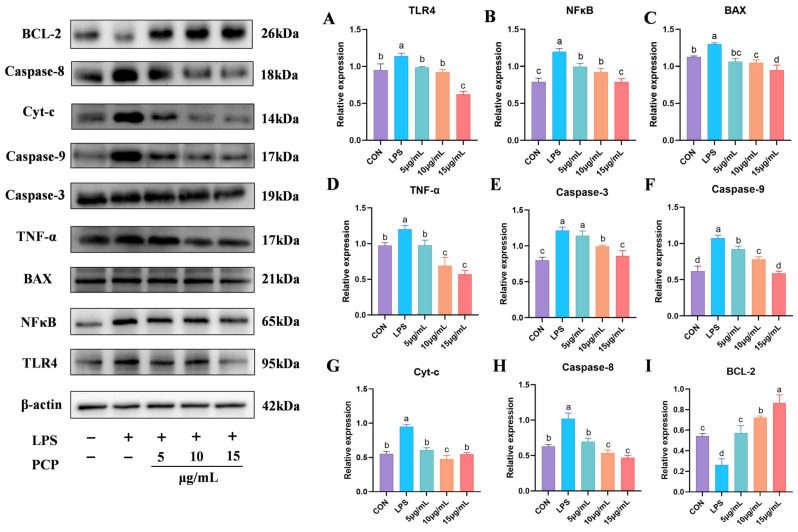
Quantification of NFκB and apoptosis signaling pathway-related proteins expression by Western blotting. (**A**) TLR4 protein expression, (**B**) NFκB protein expression, (**C**) BAX protein expression, (**D**) TNF-α protein expression, (**E**) Caspase-3 protein expression, (**F**) Caspase-9 protein expression, (**G**) Cyt-c protein expression, (**H**) Caspase-8 protein expression, (**I**) BCL-2 protein expression. All experiments were repeated more than three times and are presented as mean ± SD (significant differences (*p* < 0.05) between groups with different superscripts a, b, c, and d).

**Figure 8 cimb-47-00139-f008:**
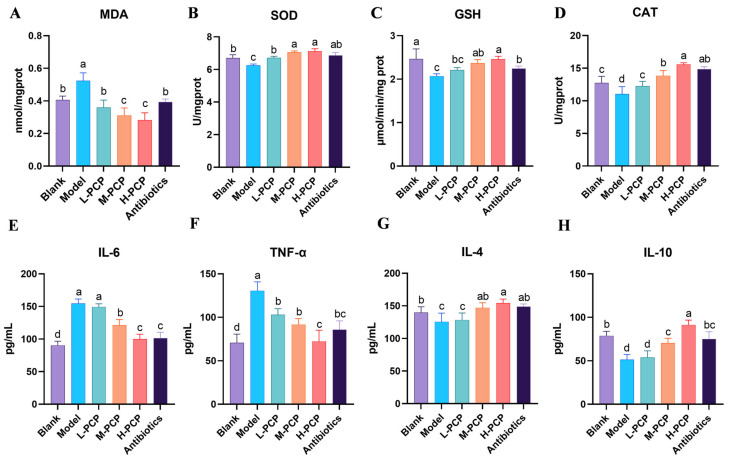
The effects of PCP on liver antioxidant and serum inflammatory factors in rats with endometritis, grouped by blank, low, medium, and high doses, and antibiotic group. (**A**) Liver MDA levels, (**B**) liver SOD levels, (**C**) liver GSH levels, (**D**) liver CAT levels, (**E**) serum IL-6 levels, (**F**) serum TNF-α levels, (**G**) serum IL-4 levels, (**H**) serum IL-10 levels. All experiments were repeated more than three times and are presented as mean ± SD (significant differences (*p* < 0.05) between groups with different superscripts a, b, c, and d).

**Figure 9 cimb-47-00139-f009:**
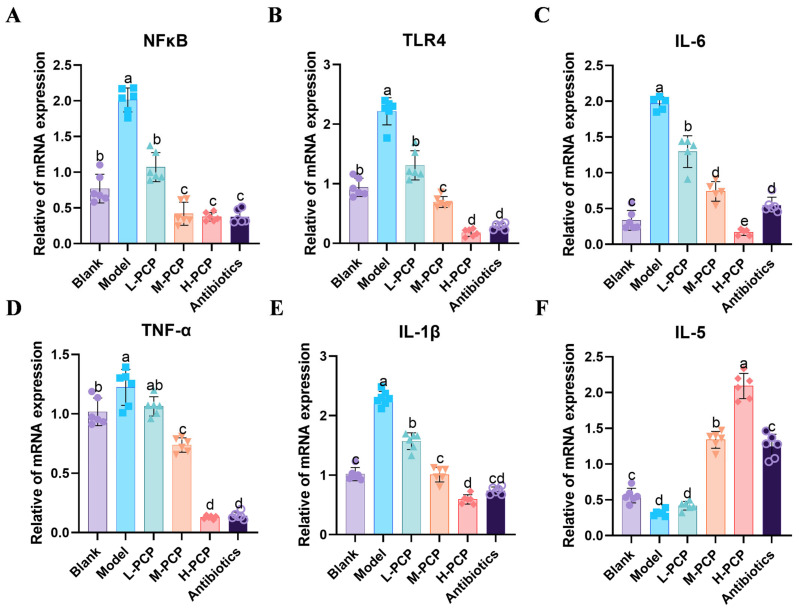
Expression levels of relevant inflammatory factor mRNA in rat uterine tissue. (**A**) *NFκB* mRNA expression, (**B**) *TLR4* mRNA expression, (**C**) *IL-6* mRNA expression, (**D**) *TNF-α* mRNA expression, (**E**) *IL-1β* mRNA expression, (**F**) *IL-5* mRNA expression. All experiments were repeated more than three times and are presented as mean ± SD (significant differences (*p* < 0.05) between groups with different superscripts a, b, c, d and e).

**Figure 10 cimb-47-00139-f010:**
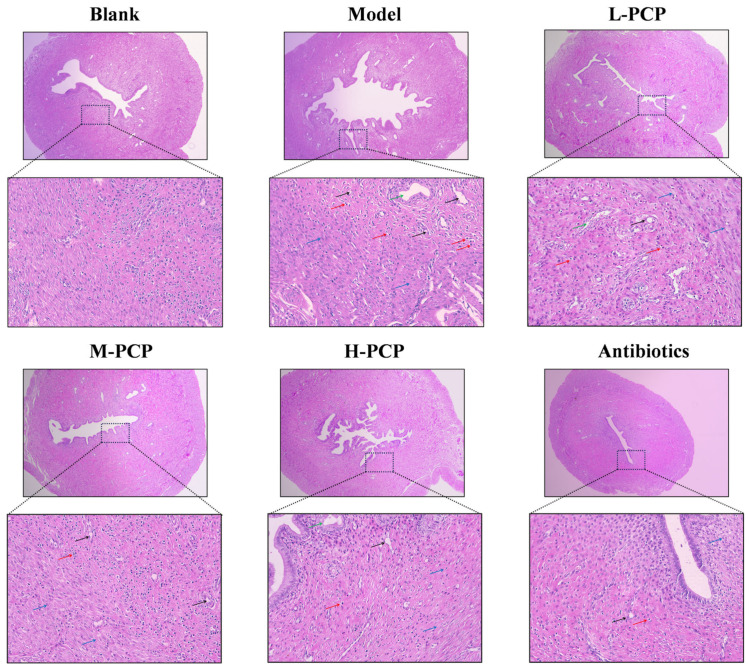
Pathological sections of rat uterine tissues after bacterial infection and intervention with PCP. The red arrows represent inflammatory cells; the black arrows represent small blood vessels; the blue arrows represent the endometrial stroma; the green arrows represent the endometrial glands.

**Table 1 cimb-47-00139-t001:** Transcriptome sequencing results.

Sample	Clean Reads Pairs	Clean Base (bp)	Length	Q20 (%)	Q30 (%)	GC (%)
Control-1	45,197,184	13,559,155,200	150	99.14	96.88	51.40
Control-2	42,640,925	12,792,277,500	150	99.07	96.63	51.34
Control-3	41,070,775	12,321,232,500	150	99.07	96.64	51.02
LPS-1	28,939,955	8,681,986,500	150	99.08	96.67	50.17
LPS-2	42,217,295	12,665,188,500	150	99.10	96.77	51.31
LPS-3	41,428,752	12,428,625,600	150	99.09	96.69	50.98

**Table 2 cimb-47-00139-t002:** Primers for real-time quantitative PCR in bovine.

Gene	Primer Sequence (5′-3′)	GenBank Accession No.
*GAPDH*	F:ATGCTGGTGCTGAGTATGTG	NC_037332.1
	R:CTTCTGGGTGGCAGTGAT	
*CDK4*	F:TTGGTGTCGGTGCCTAT	NC_037332.1
	R:CGAACGGTGCTGATGG	
*CCND1*	F:CTGGTCCTGGTGAACAAACTC	NC_037356.1
	R:CACAGAGGGCAACGAAGGT	
*CCNA2*	F:GAGTATGTCCCTGTTCCT	NC_037333.1
	R:TTGGTCCTGGTAAAGTAA	
*CDK6*	F:TGCCCACTGAAACCATAA	NC_037331.1
	R:GACCACTGAGGTAAGAGCC	
*Caspase-4*	F:GCTGCCCTTGACATCCTT	NC_037342.1
	R:CCCTGGCTGTGAGTTTCT	
*BIRC3*	F:GCCTCTTCTCAGCCTACT	NC_037342.1
	R:AGCATCATCCTTCGGTTC	
*NFκB*	F:ACACGTATCGAAGGACAGCC	NC_037348.1
	R:GTCCTCCTTCACCTCTGTGC	
*TLR4*	F:TGCCTTCACTACAGGGACTT	NC_037335.1
	R:GGGACACCACGACAATAACC	
*IL-6*	F:ATCCTGAAGCAAAAGATCGCAG	NC_037331.1
	R:TTGCGTTCTTTACCCACTCGT	
*IL-1β*	F:CGACGAGTTTCTGTGTGACG	NC_037338.1
	R:TCATGCAGAACACCACTTCTC	
*IL-10*	F:CACAGGCTGAGAACCACG	NC_037343.1
	R:CAGGGCAGAAAGCGATGA	
*BAX*	F:CAAACTGGTGCTCAAGGC	NC_037345.1
	R:GCACTCCAGCCACAAAGAT	
*FADD*	F:GGGCTTGAGGAGTGGGT	NC_037356.1
	R:GGTGAGCGTAGGCATCG	
*TRADD*	F:ACTGCCCTAGCAGAGAGTGG	NC_037345.1
	R:CTGAAACGCAGTTGCACGAT	
*Caspase-3*	F:AGTGGTGCTGAGGATGAC	NC_037354.1
	R:ACCCGAGTAAGAATGTGC	
*Caspase-8*	F:TCACCCACGGAAACAAGG	NC_037329.1
	R:TCGGTCTCAACGGCTACA	
*Caspase-9*	F:CCCTTCCTTTGTTCATCTCC	NC_037343.1
	R:TGCTTGTCTGCTGGTCTTC	
*BCL-2*	F:CATGTGTGTGGAGAGCGTCA	NC_037351.1
	R:TACAGCTCCACAAAGGCGTC	

**Table 3 cimb-47-00139-t003:** Primers for real-time quantitative PCR in rat.

Gene	Primer Sequence (5′-3′)	GenBank Accession No.
*GAPDH*	F:TTCAACGGCACAGTCAAG	NC_086022.1
	R:TACTCAGCACCAGCATCA	
*IL-6*	F:CAGAGTCATTCAGAGCAATAC	NC_086022.1
	R:GATGGTCTTGGTCCTTAGC	
*TLR4*	F:CTAGACACTTTATCCAGAGCCGTTG	NC_086023.1
	R:AAGGACAATGAAGATGATGCCAGAG	
*NFκB*	F:TGTGGTGGAGGACTTGCTGAG	NC_086020.1
	R:GCTGCCTTGCTGTTCTTGAGTAG	
*IL-1β*	F:GCAGCTTTCGACAGTGAGGAG	NC_086021.1
	R:TCTGGACAGCCCAAGTCAAG	
*TNF-α*	F:ATGGGCTCCCTCTCATCAGT	NC_086028.1
	R:GCTTGGTGGTTTGCTACGAC	
*IL-5*	F:TGACGAGCAATGAGACGATG	NC_086038.1
	R:ACTTCCATTGCCCACTCTGT	

**Table 4 cimb-47-00139-t004:** Organ indicators after rat infection and treatment with PCP.

Indicators	Group					
	Blank	Model	L-PCP	M-PCP	H-PCP	Antibiotics
Uterine index	0.20 ± 0.81 ^e^	0.42 ± 0.71 ^a^	0.32 ± 0.84 ^ab^	0.31 ± 0.76 ^bc^	0.26 ± 0.78 ^cd^	0.24 ± 0.88 ^de^
Spleen index	0.16 ± 0.82 ^c^	0.22 ± 0.80 ^a^	0.21 ± 0.89 ^a^	0.19 ± 0.81 ^ab^	0.18 ± 0.81 ^bc^	0.17 ± 0.90 ^bc^
Liver index	3.21 ± 0.77 ^b^	3.36 ± 0.18 ^a^	3.27 ± 0.65 ^a^	3.10 ± 0.56 ^a^	3.13 ± 0.58 ^ab^	3.03 ± 0.52 ^ab^

Note: All trial were repeated three times and are expressed as mean ± SD, significant differences between groups with different superscripts a, b, c, d and e (*p* < 0.05).

**Table 5 cimb-47-00139-t005:** Relative quantification of the uterus.

Item	Blank	Model	L-PCP	M-PCP	H-PCP	Antibiotics
Infiltration area of inflammatory cells (%)	0.66 ± 0.04 ^d^	26.76 ± 2.56 ^a^	25.38 ± 1.80 ^a^	15.87 ± 1.76 ^b^	13.13 ± 1.27 ^c^	15.63 ± 1.50 ^b^

Note: All experiences were repeated three times and expressed as mean ± SD, significant differences between groups with different superscripts a, b, c, and d (*p* < 0.05).

## Data Availability

Data will be made available on request.

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
