# Peer review of "From Genes to Healing: The Protective Mechanisms of Poria cocos Polysaccharide in Endometrial Health"

_cimb, 2025, doi:10.3390/cimb47030139_

Round 1
Reviewer 1 Report
Comments and Suggestions for Authors
The abstract provides a clear summary of the study. However, it lacks specificity regarding the clinical implications of PCP in bovine health. It is suggested to include a sentence discussing the translational potential of these findings to real-world veterinary applications.
In the introduction part, it is suggested to add a brief overview of prior studies on the anti-inflammatory properties and the limitations of PCP.
There are certain typographical issues, spelling mistakes and grammatical errors. It is suggested that the authors revise the manuscript thoroughly to ensure that it is free from such errors.
Was the dosage of PCP selected based on prior studies, or was it determined empirically for this research? Also, could higher doses offer additional benefits?
The authors are suggested to include the potential limitations of the PCP treatment in practical settings, if any?
I do not see any figure files in the manuscript, just the figure captions. Is that by mistake or is it an incomplete file?
Comments on the Quality of English LanguageThe english language needs to be improved. There are a lot of typographical errors, spelling mistakes and grammatical errors. It is suggested to thoroughly revise the manuscript or maybe get it revised by a native speaker to ensure that the manusript is free from such errors. This will improve the readability.
Author Response
Dear respected Reviewer:
Thanks so much for your email dated on 31-Jan-2025 regarding our manuscript “From Genes to Healing: The Protective Mechanisms of Poria Cocos Polysaccharide in Endometrial Health” (Manuscript ID: cimb-3456089). And thank you for valuable comments and suggestions on our research. Based on your comment and request, we have made modification on the original manuscript. In this revised version, changes to our manuscript within the document were all highlighted by using yellow colored text. Point-by-point responses to the reviewer are listed below this letter (see red marks). We have also submitted the revised manuscript [R1] (see yellow marks) and supplementary Material to the journal. All Authors have read the revised manuscript [R1] and agreed to submit it in its current form for consideration for publication in Current Issues in Molecular Biology.
We are glad to answer any further questions and perform further revisions if deemed necessary. Your assistance is highly appreciated.
Comment 1: The abstract provides a clear summary of the study. However, it lacks specificity regarding the clinical implications of PCP in bovine health. It is suggested to include a sentence discussing the translational potential of these findings to real-world veterinary applications.
Response: We appreciate your feedback regarding the manuscript. We have added information about the conversion potential of Poria polysaccharides in practical veterinary applications in the abstract. PCP demonstrate a high safety profile and may serve as a viable alternative treatment for clinical cases of endometritis. See Line 25-29.
Comment 2: In the introduction part, it is suggested to add a brief overview of prior studies on the anti-inflammatory properties and the limitations of PCP.
Response: Thank you for your detailed review and valuable feedback on our manuscript. This is essential for improving the manuscript. We have now added a supplementary overview in the introduction regarding the anti-inflammatory properties and limitations of Poria cocos polysaccharides. PCP possesses anti-inflammatory properties by stimulating the activation of immune cells, regulating the secretion of inflammatory factors and scavenging free radicals. Although PCP has significant anti-inflammatory and immunomodulatory effects, its limitations such as low bioavailability, complex mechanism of action, large individual variation and lack of clinical data still need to be further studied. See Line 68-73.
References:
Liu, J.; Hong, W.; Li, M.; Xiao, Y.; Yi, Y.; Liu, Y.; Wu, G. Transcriptome analysis reveals immune and metabolic regulation effects of Poria cocos polysaccharides on Bombyx mori larvae. Front Immunol 2022, 13, 1014985, doi:10.3389/fimmu.2022.1014985.
Wu, Y.; Li, D.; Wang, H.; Wan, X. Protective Effect of Poria Cocos Polysaccharides on Fecal Peritonitis-Induced Sepsis in Mice Through Inhibition of Oxidative Stress, Inflammation, Apoptosis, and Reduction of Treg Cells. Front Microbiol 2022, 13, 887949, doi:10.3389/fmicb.2022.887949.
Yang, X.; Lu, S.; Feng, Y.; Cao, C.; Zhang, Y.; Cheng, S. Characteristics and properties of a polysaccharide isolated from Wolfiporia cocos as potential dietary supplement for IBS. Front Nutr 2023, 10, 1119583, doi:10.3389/fnut.2023.1119583.
Comment 3: There are certain typographical issues, spelling mistakes and grammatical errors. It is suggested that the authors revise the manuscript thoroughly to ensure that it is free from such errors.
Response: We gratefully appreciate for your valuable comment. The issues with the manuscript printing, spelling errors, and grammatical mistakes have now been corrected and improved.
Comment 4: Was the dosage of PCP selected based on prior studies, or was it determined empirically for this research? Also, could higher doses offer additional benefits?Response: Thank you for your detailed review and valuable feedback on our manuscript. The dosage of PCP in the article was determined based on previous studies and relevant literature. The current trial is in the exploratory stage, and higher doses may have better efficacy for endometritis. However, we cannot rule out the possibility that high-dose medication may have toxic effects on animal diseases. Therefore, further investigations will be conducted in subsequent trials to determine the optimal drug dosage.
Comment 5: The authors are suggested to include the potential limitations of the PCP treatment in practical settings, if any?
Response: We gratefully thanks for the precious time the reviewer spent making constructive remarks. We have added the potential limitations of PCP treatment in real-world settings to the introduction section. See Line 70-73.
Comment 6: I do not see any figure files in the manuscript, just the figure captions. Is that by mistake or is it an incomplete file?
Response: Thank you for your detailed review and valuable comments on our manuscript. We sincerely apologize for the omission of figures in the original submission and have subsequently re-uploaded the figure files as attachments.

Reviewer 2 Report
Comments and Suggestions for Authors
Dear Authors,
First, I wanted to thank you for the chance to review your manuscript. Endometritis remains a challenging issue worldwide. The manuscript seeks to unravel the therapeutic potential of Poria cocos polysaccharides (PCP) in vitro and in vivo models. This work is well-configured and addresses a clinical problem of importance. There are, however, a number of areas in which the manuscript could be improved in terms of increasing its impact and scientific rigor. Here are the comments I have categorized as major and minor.
Major Comments
1—Importantly, the justification for targeting Poria cocos as a therapeutic agent needs to be further developed. Although the manuscript emphasizes its anti-inflammatory and antioxidant activities, a more specific comparison with other therapeutic or natural compounds widely available would improve the context of the study.
2- PCP effects, dose-dependently, are interesting and deserve further studies. The inclusion of intermediate or decreasing doses of PCP may aid in determining the lowest effective dose. Also, long-term studies of recurrence of endometritis follow-up of treatment with PCP would add great value.
3- The comparison with the antibiotics in this context is underdeveloped. The brief discussion should describe how PCP compares to antibiotics with regard to inflammation reduction, restoration of tissue structure, and prevention of recurrence.
4- Further experiments are recommended for this project or future projects to improve mechanistic understanding of PCP's impacts. Employing some techniques such as co-immunoprecipitation or molecular docking to assess the direct interaction of PCP with pivotal molecules within the TLR4/NFκB signaling cascade would bring topmost clarity. The antioxidant properties itself could be validated by measuring ROS and RNS in treated cells. Comparative time-course experiments tracking cytokine expression and apoptosis markers would better characterize the pathological progression imparted by PCP.
5- Despite its value, histology is inherently qualitative, unable to present readers with quantitative data. Image-based analysis quantifying features such as inflammatory cell infiltration and tissue damage would be more objective and rigorous. This would add to qualitative observations and improve the overall conclusions.
Minor Comments
6- Use the same legends and superscripts throughout all figures, particularly in bar charts. It is intended to make it easier for readers to interpret the results.
7- Adding annotations to highlight important pathological features in histological images would help readers visualize this information better.
8- Some cytokines such as IL-5 implicated in endometritis pathophysiology are mentioned but not further explained.
9- Expand on why apoptosis-related genes were chosen, like why some genes are specifically involved in endometritis such as BIRC3 and Caspase-4.
10- The antioxidants markers (MDA, SOD, GSH, CAT) should lead to a discussion of potential clinical relevance - what of reduced oxidative stress, the potential for tissue repair, and how this enhances translational relevance.
11- Transparency about qPCR fold-change values, with error bars in the figures, should be demonstrated to show variation and confidence in the results. Please include the qPCR efficiency of each gene. At least provide two items from the MIQE list.
12- Evaluate other natural compounds or treatments as controls to contextualize the relative effectiveness of PCP and its potential superiority over other therapies.
13- As for more mechanistic insight, although the manuscript briefly discusses apoptosis, an explanation as to how PCP affects major apoptotic proteins and pathways e.g. (BCL-2, Caspase-3) can be useful for readers.
14- The authors should clearly state which statistical methods were used to analyze the data, which tests were used to assess significance, and how potential outliers were dealt with.
There are some minor stylistic issues and not always a consistent use of tense (in the "Materials and Methods" in particular), but in general the manuscript is well written. In my opinion, the authors opted to use (imperative) language in many parts of the paper (i.e. add HPR-labeled sheep anti-rabbit IgG, and weigh the collected rat uterus) but that is not a convention in scientific writing. Although the section of the report entitled "Materials and Methods" should normally be written in the past tense, describing experiments that have already been performed (such as "HRP-labeled sheep anti-rabbit IgG was added" or "Collected rat uterus weighing"), the descriptions of the "Materials and Methods" section in this manuscript are usually written in the present tense, which is not consistent with scientific literacy. Using the imperative tense gives the impression of an instruction manual, which isn’t the right tone for reporting on scientific experiments.
The grammar and sentence structure seem to have occasional problems (such as word choice). For example, an imprecise phrase, “positive therapeutic response,” is used (lines 417–420), and a specific definition, such as “significant improvement in inflammatory markers” should be used instead. Similarly, some expressions such as “notably decreased,” and “markedly elevated” are repeated across the text and can be replaced with more varied and specific descriptions to make the text more comprehensible. Overlong sentences need to be broken into clearer statements. For example, they state the following in a long and confusing way: "Following PCP intervention, the expression of IL-5 was notably decreased compared to the control group (P<0.05)", then just rephrase the same sentence briefly "PCP treatment significantly reduced IL-5 expression compared to the control group (P<0.05)."
I believe the manuscript should be proofread and rewritten.
Author Response
Dear respected Reviewer:
Thanks so much for your email dated on 6-Feb-2025 regarding our manuscript “From Genes to Healing: The Protective Mechanisms of Poria Cocos Polysaccharide in Endometrial Health” (Manuscript ID: cimb-3456089). And thank you for valuable comments and suggestions on our research. Based on your comment and request, we have made modification on the original manuscript. In this revised version, changes to our manuscript within the document were all highlighted by using yellow colored text. Point-by-point responses to the reviewer are listed below this letter (see red marks). We have also submitted the revised manuscript [R1] (see yellow marks) and supplementary Material to the journal. All Authors have read the revised manuscript [R1] and agreed to submit it in its current form for consideration for publication in Current Issues in Molecular Biology.
We are glad to answer any further questions and perform further revisions if deemed necessary. Your assistance is highly appreciated.
Comment 1: Importantly, the justification for targeting Poria cocos as a therapeutic agent needs to be further developed. Although the manuscript emphasizes its anti-inflammatory and antioxidant activities, a more specific comparison with other therapeutic or natural compounds widely available would improve the context of the study.
Response: We appreciate your feedback regarding the manuscript. In the context of our research, we undertook a detailed comparative analysis of Poria polysaccharides in relation to other therapeutic or natural compounds. As a natural substance, Poria polysaccharides exhibit notable advantages concerning safety, versatility, and extensive applicability. Nevertheless, when juxtaposed with certain highly effective targeted pharmaceuticals, the mechanism of action of Poria polysaccharides is comparatively intricate, and there remains a need for further optimization of their bioavailability and therapeutic efficacy. See Line 59-73.
Comment 2: PCP effects, dose-dependently, are interesting and deserve further studies. The inclusion of intermediate or decreasing doses of PCP may aid in determining the lowest effective dose. Also, long-term studies of recurrence of endometritis follow-up of treatment with PCP would add great value.
Response: We appreciate your thorough evaluation and insightful comments regarding our manuscript. The present trial is in the exploratory stage, and the incorporation of moderate or reduced doses of PCP may assist in establishing the minimum effective dosage. Consequently, additional research will be undertaken in future trials to ascertain the optimal dosage of the drug. Furthermore, a long-term follow-up study will be conducted to investigate the recurrence of endometritis receiving PCP treatment.
Comment 3: The comparison with the antibiotics in this context is underdeveloped. The brief discussion should describe how PCP compares to antibiotics with regard to inflammation reduction, restoration of tissue structure, and prevention of recurrence.
Response: We gratefully appreciate for your valuable comment. In our discussion regarding antibiotics, we incorporated a comparative analysis of the role of PCP in mitigating inflammation, restoring tissue architecture, and preventing recurrence. See Line 611-617.
Comment 4: Further experiments are recommended for this project or future projects to improve mechanistic understanding of PCP's impacts. Employing some techniques such as co-immunoprecipitation or molecular docking to assess the direct interaction of PCP with pivotal molecules within the TLR4/NFκB signaling cascade would bring topmost clarity. The antioxidant properties itself could be validated by measuring ROS and RNS in treated cells. Comparative time-course experiments tracking cytokine expression and apoptosis markers would better characterize the pathological progression imparted by PCP.
Response: Thank you for your detailed review and valuable feedback on our manuscript. This study primarily investigates the effectiveness of PCP in the treatment of endometritis and is presently in the initial exploratory phase. Future experiments will involve a comprehensive analysis aimed at assessing the direct interactions of critical molecules within the PCP and TLR4/NFκB signaling pathway, as well as measuring the expression levels of ROS and RNS in cells following treatment with Poria polysaccharides.
Comment 5: Despite its value, histology is inherently qualitative, unable to present readers with quantitative data. Image-based analysis quantifying features such as inflammatory cell infiltration and tissue damage would be more objective and rigorous. This would add to qualitative observations and improve the overall conclusions.
Response: We gratefully thanks for the precious time the reviewer spent making constructive remarks. We used Image Pro-Plus (IPP) software to quantify the area of inflammatory cell infiltration, thereby facilitating a comparative analysis of the severity of inflammation among the groups. The quantitative analysis data is shown in Table â…¤. See Line 221-223 and 468.
Comment 6: Use the same legends and superscripts throughout all figures, particularly in bar charts. It is intended to make it easier for readers to interpret the results.
Response: Thank you for your detailed review and valuable comments on our manuscript. We have standardized all the legends and superscripts of the figures in the manuscript.
Comment 7: Adding annotations to highlight important pathological features in histological images would help readers visualize this information better.
Response: We gratefully appreciate for your valuable comment. We have added annotations to the important pathological features in the pathological sections of rat uterine tissue.
Comment 8: Some cytokines such as IL-5 implicated in endometritis pathophysiology are mentioned but not further explained.
Response: We gratefully thanks for the precious time the reviewer spent making constructive remarks. In the discussion of the revised manuscript, we added relevant information about the involvement of the cytokine IL-5 in the repair of endometrial tissue damage. See Line 589-595.
Comment 9: Expand on why apoptosis-related genes were chosen, like why some genes are specifically involved in endometritis such as BIRC3 and Caspase-4.
Response: Thank you for your detailed review and valuable comments on our manuscript. The selection of genes was guided by transcriptomic analyses, which indicated that genes exhibiting upregulation are predominantly associated with the apoptosis pathway, whereas those showing downregulation are primarily involved in the cell cycle pathway. Through the analysis of protein interaction networks, we identified genes that are closely linked to both apoptosis and the cell cycle. Notably, BIRC3 is integral to the apoptosis process, functioning as an anti-apoptotic factor through its interactions with tumor necrosis factor receptor-associated factors TRAF1 and TRAF2. Furthermore, the inflammatory cytokine IL-1β has been shown to enhance the expression of BIRC3, thereby augmenting cellular resistance. Conversely, the application of Caspase-4 specific siRNA effectively inhibits the production of IL-1β, thereby confirming the reliance of the endometrial cell inflammasome on activated Caspase-4. In conclusion, we have selected genes such as BIRC3 and Caspase-4 for further validation, thereby reinforcing existing research findings to strengthen the overall argument of the manuscript. In addition, we have also included a description of the transcriptome results in the discussion of the revised manuscript. See Line 560-563.
Comment 10: The antioxidants markers (MDA, SOD, GSH, CAT) should lead to a discussion of potential clinical relevance - what of reduced oxidative stress, the potential for tissue repair, and how this enhances translational relevance.
Response: Thank you for your detailed review and valuable comments on our manuscript. In the discussion section of the revised manuscript, we incorporated pertinent information regarding the effects of PCP and antibiotic intervention. Following these treatments, there was a notable reduction in the accumulation of MDA in the liver, alongside an increase in the expression levels of CAT, GSH, and SOD. These changes contributed to a decrease in oxidative stress induced by bacterial activity, thereby improving the organism's antioxidant capacity and enhancing its tissue repair mechanisms. See Line 605-610.
Comment 11: Transparency about qPCR fold-change values, with error bars in the figures, should be demonstrated to show variation and confidence in the results. Please include the qPCR efficiency of each gene. At least provide two items from the MIQE list.
Response: We gratefully thanks for the precious time the reviewer spent making constructive remarks. In accordance with your recommendations, we have enhanced and refined the graphical representation of the quantitative fluorescence results presented in the manuscript.
Comment 12: Evaluate other natural compounds or treatments as controls to contextualize the relative effectiveness of PCP and its potential superiority over other therapies.
Response: We appreciate your feedback regarding the manuscript. In the scope of our research, we performed a comprehensive comparative analysis of Poria cocos polysaccharides in relation to other therapeutic modalities. When juxtaposed with chemically synthesized pharmaceuticals, Poria cocos exhibits reduced toxicity and improved biocompatibility, rendering it appropriate for prolonged use. Additionally, in comparison to certain natural compounds, Poria cocos polysaccharides display more significant immunomodulatory and anti-tumor effects, particularly in terms of enhancing immune responses and inhibiting tumor growth. As a natural product, Poria cocos polysaccharides present considerable advantages concerning safety, multifunctionality, and extensive applicability. See Line 62-68.
Comment 13: As for more mechanistic insight, although the manuscript briefly discusses apoptosis, an explanation as to how PCP affects major apoptotic proteins and pathways e.g. (BCL-2, Caspase-3) can be useful for readers.
Response: Thank you for your detailed review and valuable comments on our manuscript. In our discussion after the manuscript revision, we supplemented the main apoptotic proteins and pathways related to the impact of PCP. PCP significantly increased the gene and protein expression of the anti-apoptotic protein BCL-2, while inhibiting the expression of the pro-apoptotic protein BAX. This resulted in an increased BCL-2/BAX ratio, which suppressed the mitochondrial-dependent apoptotic pathway. Furthermore, PCP markedly decreased the protein expression of Cyt-c and impeded the translocation of Cyt-c from the mitochondrial membrane to the cytoplasm, effectively obstructing the mitochondrial-mediated apoptotic pathway. Overall, PCP demonstrates anti-apoptotic properties through the modulation of BCL-2 family proteins, the inhibition of the Caspase cascade, the reduction of Cyt-c release, and the downregulation of upstream apoptosis regulatory genes. See Line 560-563 and 571-576.
Comment 14: The authors should clearly state which statistical methods were used to analyze the data, which tests were used to assess significance, and how potential outliers were dealt with.
Response: We appreciate your feedback regarding the manuscript. We have improved the information related to the statistical analysis methods for the manuscript data. A one-way ANOVA was performed utilizing SPSS software, with Duncan's multiple range test employed to evaluate the significance of differences among means. The statistical significance of these differences is denoted by the presence of letters a-d within a column; distinct letters signify significant differences between means (P<0.05). Data visualization was executed using GraphPad Prism 9.5, and data measurements were carried out with Image J software. All experimental data were independently replicated a minimum of three times, and results are presented as mean ± standard deviation (SD). See Line 225-232.

Round 2
Reviewer 1 Report
Comments and Suggestions for Authors
The revised version of the manuscript presents notable improvements over the initial submission, particularly in language clarity, structural coherence, and biological mechanism discussion. While the revised version demonstrates progress, several concerns remain unaddressed. These aspects must be addressed before the manuscript can be accepted for publication.
Consider citing previous studies that successfully translated rat model findings to bovine clinical applications.
The revised version has improved readability, but some grammatical errors and awkward phrasing remain.
"KEEG" should be corrected to "KEGG" throughout the manuscript.
"An the endometritic rat model" should be corrected to "an endometritic rat model".
Ensure consistent terminology for “differentially expressed genes (DEGs)” and “signaling pathways”.
Comments on the Quality of English LanguageCertain grammatical errors are still present in the revised form of the manuscript. Consider rechecking the manuscript thoroughly to ensure that the manuscript is free of such errors.
Author Response
Dear respected Reviewer:
Thanks so much for your email dated on 17-Feb-2025 regarding our manuscript “From Genes to Healing: The Protective Mechanisms of Poria Cocos Polysaccharide in Endometrial Health” (Manuscript ID: cimb-3456089). And thank you for valuable comments and suggestions on our research. Based on your comment and request, we have made modification on the original manuscript. In this revised version, changes to our manuscript within the document were all highlighted by using yellow colored text. Point-by-point responses to the reviewer are listed below this letter (see red marks). We have also submitted the revised manuscript [R1] (see yellow marks) and supplementary Material to the journal. All Authors have read the revised manuscript [R1] and agreed to submit it in its current form for consideration for publication in Current Issues in Molecular Biology.
We are glad to answer any further questions and perform further revisions if deemed necessary. Your assistance is highly appreciated.
Comment 1: Consider citing previous studies that successfully translated rat model findings to bovine clinical applications.
Response: Thank you for your detailed review and valuable feedback on our manuscript. In the introduction section, we added feasibility study on the use of rat models for researching bovine endometritis. The study elucidated the potential mechanisms of drug treatment for endometritis through LPS-induced bovine endometrial epithelial cells and rat endometritis models. See Line 81-84.
References:
Shen, W.; Oladejo, A.O.; Ma, X.; Jiang, W.; Zheng, J.; Imam, B.H.; Wang, S.; Wu, X.; Ding, X.; Ma, B.; et al. Inhibition of Neutrophil Extracellular Traps Formation by Cl-Amidine Alleviates Lipopolysaccharide-Induced Endometritis and Uterine Tissue Damage. Animals (Basel) 2022, 12, doi:10.3390/ani12091151.
Wu, H.; Dai, A.; Chen, X.; Yang, X.; Li, X.; Huang, C.; Jiang, K.; Deng, G. Leonurine ameliorates the inflammatory responses in lipopolysaccharide-induced endometritis. Int Immunopharmacology 2018, 61, 156-161, doi: 10.1016/j.intimp.2018.06.002.
Comment 2: The revised version has improved readability, but some grammatical errors and awkward phrasing remain.
Response: We gratefully appreciate for your valuable comment. The issues with the spelling errors, and grammatical mistakes have now been corrected and improved.
Comment 3: "KEEG" should be corrected to "KEGG" throughout the manuscript.
Response: Thank you for your detailed review and valuable feedback on our manuscript. The spelling errors in the manuscript have been rectified and enhanced. See Line 12, 291 and 301.
Comment 4: "An the endometritic rat model" should be corrected to "an endometritic rat model".
Response: We gratefully thanks for the precious time the reviewer spent making constructive remarks. The issues with the grammatical mistakes have now been corrected and improved. See Line 15.
Comment 5: Ensure consistent terminology for “differentially expressed genes (DEGs)” and “signaling pathways”.
Response: We gratefully thanks for the precious time the reviewer spent making constructive remarks. We have standardized and corrected the descriptive terms in the manuscript.

Reviewer 2 Report
Comments and Suggestions for Authors
Dear Authors,
Thanks for addressing the comments.
All the best,
Author Response
Dear Reviewer :
I sincerely thank you for your comments and support, which provide us with greater motivation to move forward.
Your professional advice is of great significance to me, and I feel very honored and grateful for it. I will carefully consider your suggestions, and the subsequent improvements to the paper will further reflect the value of your guidance.
Best wishes to you!